# Prevalence of anemia and its associated factors among under-five children living in Arba Minch Health and Demographic Surveillance System Sites (HDSS), Southern Ethiopia

**Samuel Alemu Bamboro**[1]*, **Hape Ibren Boba**[2], **Mihiret Kitaw Geberetsadik**[1], **Zeleke Gebru**[1], **Befikadu Tariku Gutema**[1]

**1** School of Public Health, College of Medicine and Health Sciences, Arba Minch University, Arba Minch, Ethiopia, **2** Teltele Woreda Health Office, Borena Zone, Oromia Regional State, Ethiopia

* bizuneshhaile23@gmail.com, samuel.alemu@amu.edu.et

**Data Availability Statement:** Data used in this analysis are available at https://osf.io/ze7pk.

## Abstract

Childhood anemia affects around half of under five children and has impacts on physical, mental, and social development, both in the short and long term. The objective of the study was to determine the prevalence of anemia and its associated factors among under-five children living in Arba Minch Health and Demographic Surveillance System Sites (HDSS). A community-based cross-sectional study was conducted among randomly selected under-five children with their caregivers living in Arba Minch HDSS from June to August 2023. The questionnaire was developed to assess sociodemographic, nutrition, healthcare, and environmental characteristics. Hemoglobin concentration was adjusted for altitude of the village. Anemia was defined as the hemoglobin concentration below 11 g/dL. It was further categorized as mild (between 10–10.9 g/dL), moderate (7–9.9 g/dL), and severe (less than 7 g/dL). The analysis of factors associated with anemia was assessed by using logistic regression and significance was determined at p-value <0.05. A total of 332 under-five children with a mean (SD) age of 33(15) months participated. The overall prevalence of anemia among under-five children was 35.3% (95% CI: 30.4, 40.7). The magnitude of mild, moderate, and severe anemia was 12.4% (95%CI: 9.2, 16.4), 22.1% (95%CI: 17.9, 26.9), and 0.9% (95% CI: 0.3, 2.8), respectively. Anemia had a negative association with the advancing age of the children (0.95, 95%CI: 0.93, 0.97) and improvement in the family wealth score (0.86, 95% CI: 0.75, 0.99). Nearly one-third of the under-five children were anemic and childhood anemia is a moderate public health problem in the study area. The magnitude of anemia was negatively associated with the advance in child age and with the increase in the wealth status of the household. It is important to provide due attention to reduce the magnitude of anemia specifically for the youngest children and for those children from poor households.

**Funding:** The work was supported by Arba Minch University (in the context of Master of Public Health study of SAB). The funder had no role in study design, data collection and analysis, decision to publish, or preparation of the manuscript.

**Competing interests:** The authors have declared that no competing interests exist.

## Introduction

Anemia is a blood disorder characterized by insufficient RBCs, and a decreased capacity to transport oxygen which is ultimately insufficient to fulfill the body's physiological demands. Anemia can be classified as mild (between 10–10.9 g/dL), moderate (7–9.9 g/dL), and severe (less than 7 g/dL) using the hemoglobin concentration [1, 2]. Anemia affects the growth, development, and energy of children adversely [2]. Nutritional deficiencies, infection, and genetics are among the causes of anemia. Iron deficiency accounts for nearly half of all cases in children [3]. Additional nutritional deficiency causes of anemia include folate, vitamin B12, and, vitamin A deficiencies. Malaria and intestinal worms (hookworm and Schistosoma infections) are additional causes for the development of anemia among children specifically in developing countries [3, 4].

Childhood anemia increases the risk of mortality and morbidity in children which is linked to a decreased capacity to fight infections [5, 6]. In addition, anemia during childhood has a negative impact on the social, physical, and mental development of children in both the short and long term. It may cause immune function abnormalities, poor motor and cognitive development, poor school performance, and reduced work productivity in children, lowering earning potentials during adulthood and subsequently followed by a negative effect on countries national economic growth [7, 8]. Severe anemia has a significant morbidity and fatality rate and it is the chief cause or contributing factor for overall under-five mortality [9].

Anemia affected more than 1.8 billion people and caused 50.3 million years to live with disabilities in 2021. This massive burden represented 5.7% of all people living with disabilities [3]. The World Health Organization (WHO) also reported that 42% of children under-five are anemic [2]. Africa accounts for 67.6% of these children with anemia [1, 10]. According to the Ethiopian Demographic and Health Survey (EDHS) 2016 report, 57% of children aged 6–59 months were anemic [11]. The prevalence significantly decreased as per the national food and nutrition strategy baseline survey report of 2023, which was 16% [12]. Studies also indicated child's age and sex, place of residence, nutritional status, feeding practices, and socioeconomic background are associated with anemia. Anemia is more prevalent among younger children, female, those who are undernourished (experiencing stunting, wasting, or thinness) children who start complementary feeding early, those living in rural area, and those with low maternal socioeconomic status [13–17]

Moreover, there is a huge difference between anemia prevalence among pocket studies conducted in different places, and the factors associated with childhood anemia are also variable between studies from Ethiopia [18]. There is also an enormous variation of prevalence between EDHS 2005 (54%), EDHS 2011 (44%), and EDHS 2016 (57%). The recently conducted food and nutrition baseline survey of Ethiopia shows a significant decline in the prevalence of anemia which was 16% among under-five children [11, 12, 19, 20]. Therefore, we sought to see the situation in the local area along with the contextual risk factors. Thus, this study aimed to determine the prevalence of anemia and its associated factors among children aged 6–59 months residing in Arba Minch Health and Demographic Surveillance Site (HDSS), in southern Ethiopia.

## Materials and method

### Study area

The study was conducted in Arba Minch HDSS, located in Arba Minch Zuria and Gacho Baba districts of Gamo Zone, Sothern Ethiopia. Arba Minch HDSS collects demographic data and updates it every six months. From nine kebeles (the smallest administrative unit in Ethiopia)

of the HDSS, eight were included in this study (one of the kebele was excluded due to security reason during data collection period). Based on the agro-ecology of Ethiopia, Arba Minch HDSS included lowland (4 kebeles), midland (2 kebeles), and highland (3 kebeles) (the excluded kebele was from lowland). According to the 2023 report from the HDSS, the total population was 87,543.

## Study design, period and population

A community-based cross-sectional study was conducted from June to August 2023. Children aged from 6 to 59 months were included in the study. As of the latest update to the HDSS database in March 2023, a total of 8,555 children aged 6–59 months within the Arba Minch HDSS kebeles were eligible for the study. To use the updated dataset for random selection of the children, under-five children who have lived for 6 or more months in Arba Mich HDSS were included in this study. We based our sampling frame on the dataset updated 6 months before March 2023. Those selected children who were severely ill and admitted to health care facilities during data collection were excluded,. The sample size was determined using a single population proportion formula and calculated with the consideration of anemia prevalence of 15% [12], 95% confidence level, and 4% margin of error. The estimated final sample size was 337 after adding 10% of the non-response rate. The study participants were selected by using the Arba Minch HDSS database, which contains the list of all individuals in the kebeles with their birth date, and individual and household identification, as a sampling frame. Considering the number of under-five children living in each kebele, study participants were proportionally allocated to the eight kebeles. Then using Stata 14 software, the study participants were selected randomly from each kebele.

## Data collection procedure and instruments

Data was collected via a questionnaire administered by interviewees, prepared in English, and translated into Amharic language. The questionnaire was pre-tested before actual data collection on 5% (17 children) of the sample size in Chano Mile kebele (which is outside the study area), to verify the consistency of the tool and help to familiarize the data collectors with the tool. Corrections were made to keep the questionnaire clear and consistent before actual data collection. The questionnaire includes socio-demographic characteristics of the child and mother which include the age and sex of the child, education, and occupation of the mother. Household assets, building materials, access to water, availability of latrine, and number of members of the household were included regarding the household characteristics. Obstetric history regarding the indexed child and history of recent illness of the child (within the last 2 weeks) were also included. Further, the cause of recent illness was asked if the mother/caregiver went to health facility and/or knew the diagnosis.

The frequencies of consumption of over 67 locally available food ingredients for a week before the data collection were recorded for the assessment of dietary diversity score. The list of food items was developed based on an extensive interview with the data collectors, who are from the study area. Caregivers of the children were asked to report the frequency of consumption of each food item within the past week. The Household Food Insecurity Access Scale (HFIAS) which was developed by the Food and Nutrition Technical Assistance Project (FANTA) was used to assess household food security status [21]. Data for determining the household wealth score was collected by asking for ownership of selected assets that are common in the local area based on the Ethiopian Demographic and Health Survey (EDHS) 2016 wealth index variables [11].

In addition to the questionnaire, a finger prick blood sample was collected from the child and hemoglobin concentration was determined using the HemoCue (HemoCue Hb 301) at

the data collection point which was the home of the children. The data was collected by nine trained healthcare professionals. The questionnaire was uploaded on KOBO collect survey, a mobile application that allows for collecting data using mobile devices.

## Data quality assurance and control

The questionnaire was prepared in English, translated into the local language (Amharic), and then translated back into English to ensure consistency. Data collectors were trained on the study objective, consent procedures, and data collection techniques. One of the investigators (SAB) closely monitored the data collection process with the server and onsite supervision. Measurements were taken according to the manufacturer's recommended standards.

## Data processing and analysis

The data was downloaded from KOBO server in CSV format and imported to STATA 14 software for analysis. Proportions, standard deviations, and means were used to describe independent variables and anemia in the study population. Household's food security status was determined using the household food insecurity access prevalence scale and those households with mild, moderate and severe food insecure were considered as food insecure [21]. The food frequency list was aggregated into seven groups to estimate the child's dietary diversity score [22]. In addition, meat, fish and poultry, milk and milk products, and fruit and vegetables were considered separately based on their association with iron intake. The household wealth score was computed using principal component analysis (PCA) based on household assets, building materials of the house, access to water and availability of latrine. The hemoglobin concentration of the children was adjusted for altitude using the following formula: Hemoglobin correction $= -0.032$ (altitude in meter $\times 0.0032808$) $+ 0.022$ (altitude meter $\times 0.0032808)^2$ [2, 23]. Anemia was determined if the hemoglobin concentration was less than 11 g/dL. Further mild, moderate and severe anemia was determined if the hemoglobin concentration were between 10–10.9 g/dL, 7–9.9 g/dL and less than 7 g/dL, respectively [2]. A binary logistic regression model was used for data analysis, bivariable logistic regression was done using independent variables to identify variables candidate for multivariable logistic regression. In the bivariable analysis, variables with a p-value of $< 0.25$ were fitted into the multivariable analysis. Finally, factors associated with anemia were determined among the selected candidate variables using a multivariable binary logistic regression model. Multicollinearity was assessed by examining the Variance Inflation Factor (VIF) and tolerance to ensure there was no correlation between the independent variables. The final model's fitness was assessed using the Hosmer- Lemeshow test, with a p-value greater than 0.05 indicating good fit of the model to the dataset. To show the strength of the association, both the crude odds ratio (COR) and the adjusted odds ratio (AOR) with the corresponding 95% confidence interval (CI) were calculated. The statistical significance was determined in the multivariable analysis at a confidence interval of 95% at p-value $< 0.05$.

## Ethical approval and consent

Ethical clearance was obtained from the Arba Minch University Institutional Research Ethics Review Board (SA1449). Official letter of cooperation was written to Arba Minch Zuria district and Gacho Baba districts, and Kebele administrative offices from Arba Minch University School of Public Health. Finally, as the study is on minors, written informed consent was obtained from the caregivers at the time of data collection. For children with moderate to severe anemia, the data collectors referred the caregiver to a nearby health facility for further analysis and treatment.

## Results

### Socio-demographic and economic characteristics of study participants

Three hundred thirty-two under-five children with their mothers/caregivers participated in the study which makes the response rate of 98.52%. The mean (SD) age of children was 32.69 (15.12) months. Half (48.8%) of the children were male and the majority (92.8%) of the households were headed by males. Regarding the educational status of caregivers, more than half (56.3%) of mothers/caregivers did not attend formal education. The mean (SD) family size of the household was 5.53 (1.66) and half (48.8%) had more than six members. Among the households in the study area, 88% were food secure. Over two-thirds (71.4%) of the households in the study area use tap water and the majority of the households (78.3%) had a waste disposal system. Almost all the households (95.8%) had toilets which are traditional pit latrines (Table 1).

**Table 1. Characteristics of the children, caregivers and households where under-five children were living in Arba Minch HDSS, Southern Ethiopia.**

| Variable | Category | Non-anemic Frq (%) | Anemic Frq (%) | Total Frq (%) |
|---|---|---|---|---|
| Age of the children (Months) | 6–11 | 6 (2.8) | 13 (11.1) | 19 (5.7) |
| | 12–23 | 30 (14) | 48 (41) | 78 (23.5) |
| | 24–59 | 179 (83.3) | 56 (47.9) | 235 (70.8) |
| Sex of the child | Male | 109 (50.7) | 53 (45.3) | 162 (48.8) |
| | Female | 106 (49.3) | 64 (54.7) | 170 (51.2) |
| Recent illness history | No | 189 (87.9) | 101 (86.3) | 290 (87.3) |
| | Yes | 26 (12.1) | 16 (13.7) | 42 (12.7) |
| Education status of mother | No formal education | 125 (58.1) | 62 (53) | 187 (56.3) |
| | Primary education | 74 (34.4) | 47 (40.2) | 121 (36.4) |
| | 2° education & above | 16 (7.4) | 8 (6.8) | 24 (7.2) |
| Current marital status of mother | Married | 207 (96.3) | 107 (91.5) | 314 (94.6) |
| | Other$^\infty$ | 8 (3.7) | 10 (8.5) | 12 (3.6) |
| Occupation of mother | Housewife | 161 (74.9) | 80 (68.4) | 241 (72.6) |
| | Private employee/business | 46 (21.4) | 33 (28.2) | 79 (23.8) |
| | Government employee | 8 (3.7) | 4 (3.4) | 12 (3.6) |
| Family size | Five or less | 104 (48.4) | 66 (56.4) | 170 (51.2) |
| | More than five | 111 (51.6) | 51 (43.6) | 162 (48.8) |
| Number of under-five children | Participating child | 106 (49.3) | 60 (51.3) | 166 (50) |
| | Additional child | 109 (50.7) | 57 (48.7) | 166 (50) |
| Household food security status | Food secured | 195 (90.7) | 97 (82.9) | 292 (88) |
| | Food insecure | 20 (9.3) | 20 (17.1) | 40 (12) |
| Residence | Rural | 141 (65.6) | 78 (66.7) | 219 (66) |
| | Semi-urban | 74 (34.4) | 39 (33.3) | 113 (34) |
| Source of drinking water | clean tap water | 153 (71.2) | 84 (71.8) | 237 (71.4) |
| | Other sources$^a$ | 62 (28.8) | 33 (28.2) | 95 (28.6) |
| Waste disposal system | Appropriate$^\emptyset$ | 168 (78.1) | 92 (78.6) | 260 (78.3) |
| | Inappropriate$^9$ | 47 (21.9) | 25 (21.4) | 72 (21.7) |
| Latrine access | Yes | 208 (96.7) | 110 (94) | 318 (95.8) |
| | No | 7 (3.3) | 7 (6) | 14 (4.2) |

$^\infty$Other: Singel, Divorced and Widowed

$^a$Other sources: stream, river, pond

$^\emptyset$Appropriate: burning, pit, garbage can

$^9$ Inappropriate: open field, no disposal system

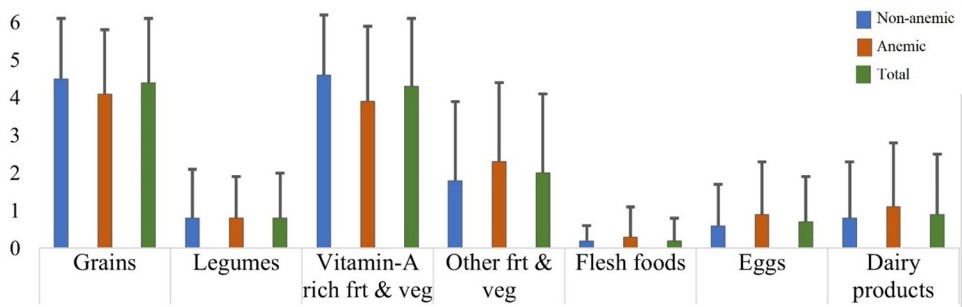

**Fig 1. Under five children average frequency of food group consumption per week by anemia status at Arba Minch HDSS (grains: Grains, white starchy roots & tubers; vitamin-A rich frt & veg: Vitamin-A rich fruits and vegetables; other frt & veg: Other fruits and vegetables; flesh foods: Meat, fish, poultry, organ meats).**

## Feeding practice and healthcare-related factors of the child and households

Out of 332 under-five children enrolled in the study, 42(12.7%) had a history of illness in the last 2 weeks (Table 1). Among those, 13(31%), 15(35.7%), 4(9.5%), and 37(88.1%) children had a history of malaria, diarrhea, intestinal parasites, and fever respectively. Regarding ANC

**Table 2. Logistic regression analysis showing factors associated with anemia among under-five children (n = 332), living in Arba Minch HDSS, Southern Ethiopia.**

| Variables | | Anemia status | | COR (95%CI) | AOR (95%CI) |
|---|---|---|---|---|---|
| | | **Normal** | **Anemic** | | |
| Age (in months) | | 36.53 (14.10) | 25.62 (14.40) | 0.95 (0.93, 0.96)[1] | 0.95 (0.93, 0.97)[1] |
| Occupation of the caregiver | Housewife | 161 (66.80) | 80 (33.20) | Ref. | |
| | Working outside[3] | 54 (59.34) | 37 (40.66) | 1.38 (0.84, 2.27) | 1.04 (0.54, 2.01) |
| Marital status of the caregiver | Married | 207 (65.92) | 107 (34.08) | Ref. | |
| | Other[4] | 8 (44.44) | 10 (55.56) | 2.42 (0.93, 6.31) | 1.62 (0.51, 5.13) |
| Family size | Five or less | 104 (61.18) | 66 (38.82) | Ref. | |
| | More than five | 111 (68.52) | 51 (31.48) | 0.72 (0.46, 1.14) | 0.79 (0.47, 1.32) |
| Latrine availability | No | 7 (50.00) | 7 (50.00) | Ref. | |
| | Yes | 208 (65.41) | 110 (34.59) | 0.53 (0.18, 1.55) | 0.7 (0.22, 2.23) |
| Household head sex | Male | 203 (65.91) | 105 (34.09) | Ref. | |
| | Female | 12 (50.00) | 12 (50.00) | 1.93 (0.84, 4.45) | 1.62 (0.58, 4.50) |
| Place delivery of index child | Health facility | 131 (61.79) | 81 (38.21) | Ref. | |
| | Home | 84 (70.00) | 36 (30.00) | 0.69 (0.43, 1.12) | 1.66 (0.78, 3.53) |
| Wealth score | | 0.21 (2.56) | -0.39 (2.83) | 0.92 (0.85, 1.00)[2] | 0.86 (0.75, 0.99)[1] |
| Food security status | Food Secure | 195 (66.78) | 97 (33.22) | Ref. | |
| | Food Insecure | 20 (50.00) | 20 (50.00) | 2.01 (1.03, 3.91)[1] | 2.47 (0.92, 6.64)[2] |
| | Meat, Fish & poultry[5] | 0.18 (0.45) | 0.29 (0.79) | 1.38 (0.92, 2.07) | 0.89 (0.58, 1.38) |
| | Milk and milk product[s] | 0.80 (1.53) | 1.10 (1.67) | 1.12 (0.98, 1.29) | 1.07 (0.91, 1.27) |
| | Fruit & vegetable[7] | 4.69 (1.63) | 4.07 (1.96) | 0.82 (0.72, 0.93)[1] | 0.93 (0.80, 1.08) |

[1]p-value <0.05

[2]p-value <0.10

[3]Working outside: Had occupation other than housewife including employee of organizations

[4]Other: divorced, single and widowed

[5]weekly frequency of Meat, Fish & poultry consumption

[6]weekly frequency of Milk and milk product consumption

[7]weekly frequency of fruit & vegetable consumption; COR: crude odds ratio; AOR: Adjusted Odds Ratio; CI: Confidence Interval.

follow-up, 293(88.3%) mothers had had ANC follow-up during pregnancy and more than half of the mothers (63.9%) gave birth to the index child at the health facility. Grains, white starchy roots & tubers, followed by vitamin-A rich fruits and vegetables were the most frequently consumed food groups while flesh food and eggs were the least (Fig 1)

### Prevalence of anemia

The mean ± SD hemoglobin concentration of the children was 11.5±1.99 g/dL and the overall prevalence of anemia among the study participants was 35.3% (95% CI: 30.4, 40.7). The magnitude of mild, moderate, and severe anemia was 12.4% (95%CI: 9.2, 16.4), 22.1% (95%CI: 17.9, 26.9), and 0.9% (95%CI: 0.3, 2.8), respectively.

### Factors associated with anemia among children under five years of age

Based on bivariate analysis, the age of the child, household food security status, and frequency of fruit and vegetable consumption per week were significantly associated with the prevalence of anemia among under-five children. The odds of anemia decrease with the child's age and the frequency of fruit and vegetables consumption, and are higher among children from food-insecure households (Table 2).

In multivariate analysis, the age of the child and the wealth score of the household are significantly associated with the prevalence of anemia. The prevalence of anemia was negatively associated with both advancing child age and higher household wealth scores, while household food security status showed a marginal association. The likelihood of anemia decreased by 0.95 (95%CI: 0.93, 0.97) with an advancing in the age of the child in months. Similarly, a unit increase in wealth score of the household decreased anemia prevalence with the odds of 0.86 (95%CI: 0.75, 0.99) among children (Table 2).

## Discussions

More than one-third (35%) of under-five children in the Arba Minch HDSS were anemic, which is regarded as a moderate public health problem [2]. The finding of the meta-analysis from Ethiopia showed that the prevalence of anemia among under-five children was around 45% with significant discrepancy among regions [13, 16]. The result of the 2016 EDHS indicated that the prevalence of anemia among under-five children was 57% while the 2023 report of the Ethiopian Public Health Institute showed only 16% of the children had anemia [11, 12]. Under-five children are the most vulnerable group of the population for malnutrition including anemia [3].

The finding showed that there is a negative association between the advancing age of the children and the prevalence of anemia; the likelihood of anemia decreased with the advancing age of the child. This is in line with findings from other part of Ethiopia [13, 24–26]. The reasons for a higher likelihood of being anemic in younger children may be related to different factors. Children's growth is relatively rapid during younger age which increases the iron and other nutrient demand. For most of the children from low-income countries including Ethiopia, the increased nutritional demand may not be replaced by the intake, which predisposes the child to anemia and other nutrition-related deficiencies [27]. In Ethiopia, children commonly consume whole cow's milk at a young age [28, 29]. Even if this finding did not confirm it, the common feeding practices of younger children, including cow and human milk, also increase the likelihood of anemia [27, 30].

This study has also found a negative association between anemia among under-five children and household economic status. The prevalence of anemia among children from households with higher wealth status was lower compared to those from lower economic status. This

finding is supported by studies conducted in Ethiopia [15, 24, 25, 31]. The UNICEF framework for malnutrition indicates that financial resource, including household wealth status, are enabling factors for accessing an adequate diet and proper health care [32]. Reports also showed the existence of a two-way link between poverty and malnutrition [33]. Moreover, the association could also be attributed to the nature of the food consumed by poor and wealthy households. Children from poor households mainly obtain iron from non-haem iron sources (plant-based sources of iron), which have low bioavailability [28, 34].

This study assessed the prevalence of anemia, and associated factors among preschool children in their community, which provide evidence in its natural set-up. For determining anemia, hemoglobin concentration was adjusted for the village altitude. Although the main cause of anemia is nutritional deficiencies, such as iron, folate, vitamin B12, and vitamin A, non-nutritional factors, such as infections and inflammation, are also known contributors to the prevalence of anemia among children. While assessing the hemoglobin concentration, we did not test for malaria and intestinal parasites. The cross-sectional design for determining the acute nutritional deficiencies and recall-based evaluation of some of the variables like dietary intake, illness, and food security may have recall bias.

## Conclusions

The results of this study indicated that anemia is a moderate public health problem among preschool children in Arba Minch HDSS. The prevalence of anemia was higher among younger children and children from lower economic status. Special attention regarding anemia should be given to younger preschool children, as this is a critical period for all nutrition and health-care-related activities for adequate growth and development of children.

## Supporting information

**S1 Text. Questionnaire.**
(DOCX)

## Acknowledgments

We would like to thank the data collectors and mothers/caregivers who participated in the study. We would also like to thank the center of Arba Minch HDSS and Aliazer Behiru (data manager of the center) for facilitating the random selection of the children from the sampling frame. We would finally like to extend our gratitude to Arba Minch Hospital Clinical Trial Centre for providing HemoCue 301 machine, Micro cuvettes, and lancets.

## Author Contributions

**Conceptualization:** Samuel Alemu Bamboro, Zeleke Gebru, Befikadu Tariku Gutema.

**Data curation:** Samuel Alemu Bamboro.

**Formal analysis:** Samuel Alemu Bamboro, Befikadu Tariku Gutema.

**Funding acquisition:** Samuel Alemu Bamboro.

**Investigation:** Samuel Alemu Bamboro, Hape Ibren Boba, Mihiret Kitaw Geberetsadik, Zeleke Gebru, Befikadu Tariku Gutema.

**Methodology:** Samuel Alemu Bamboro, Hape Ibren Boba, Mihiret Kitaw Geberetsadik, Zeleke Gebru, Befikadu Tariku Gutema.

**Project administration:** Samuel Alemu Bamboro.

**Resources:** Samuel Alemu Bamboro.

**Software:** Samuel Alemu Bamboro.

**Supervision:** Samuel Alemu Bamboro.

**Validation:** Samuel Alemu Bamboro, Zeleke Gebru, Befikadu Tariku Gutema.

**Visualization:** Samuel Alemu Bamboro, Zeleke Gebru, Befikadu Tariku Gutema.

**Writing – original draft:** Samuel Alemu Bamboro, Befikadu Tariku Gutema.

**Writing – review & editing:** Samuel Alemu Bamboro, Hape Ibren Boba, Mihiret Kitaw Geber-etsadik, Zeleke Gebru, Befikadu Tariku Gutema.

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
