## [Decision Letter · Decision Letter 0]

23 Aug 2024

PGPH-D-24-01672

Prevalence of anemia and its associated factors among under-five children living in Arba Minch Health and Demographic Surveillance System Sites (HDSS), Southern Ethiopia

Dear Dr. Bamboro,

Thank you for submitting your manuscript to PLOS Global Public Health. After careful consideration, we feel that it has merit but does not fully meet PLOS Global Public Health’s publication criteria as it currently stands. Therefore, we invite you to submit a revised version of the manuscript that addresses the points raised during the review process.

We look forward to receiving your revised manuscript.

Kind regards,

Kamala Thriemer

Academic Editor

Additional Editor Comments (if provided):

Reviewers' comments:

Reviewer's Responses to Questions

**Comments to the Author**

1. Does this manuscript meet PLOS Global Public Health’s publication criteria? Is the manuscript technically sound, and do the data support the conclusions? The manuscript must describe methodologically and ethically rigorous research with conclusions that are appropriately drawn based on the data presented.

Reviewer #1: Yes

Reviewer #2: Partly

2. Has the statistical analysis been performed appropriately and rigorously?

Reviewer #1: Yes

Reviewer #2: Yes

3. Have the authors made all data underlying the findings in their manuscript fully available (please refer to the Data Availability Statement at the start of the manuscript PDF file)?

Reviewer #1: Yes

Reviewer #2: No

4. Is the manuscript presented in an intelligible fashion and written in standard English?

Reviewer #1: Yes

Reviewer #2: No

5. Review Comments to the Author

Reviewer #1: This paper presents the results of a cross-sectional survey of anaemia prevalence and associated socio-economic and demographic risk factors in children under 5 years residing in one of 9 kebeles in Arba Minch District in Ethiopia. The study is reported clearly. The sampling frame was appropriate. Data collection procedures were robust (including forward and back translation between English and Amharic) and the analysis methods are satisfactory. The survey showed that anaemia (haemoglobin <11g/dL) was present in 35.3% of children and was negatively associated with age and recent consumption of fruits and vegetables and positively associated with increasing food insecurity. Although valid, these results are not surprising or particularly novel. The study would have been more useful if the authors had also assessed the prevalence of several other very important risk factors such as malaria, helminth infection, specific nutritional deficiencies and red cell and haemoglobin disorders.

My suggestions to the authors are all minor:

1. It is important that the authors state what the definitions of anaemia, moderate and severe anaemia are in the abstract

2. Throughout the paper, including the abstract, the authors should specify the directions of the associations found between anaemia and the socioeconomic and demographic risk factors – eg “a negative association with advancing age”

3. In the results section, age should be presented with range as well as standard deviation

4. In table 1, for consistency, suggest presenting percentages to the same number of decimal places even though it is clear that these are whole numbers

Reviewer #2: Page 2 (Introduction)

In the Abstract (conclusions), the phrase “Improvement of wealth status” seems inappropriate as the study has tackled wealth status, but not improvement of wealth status.

The given definition of anemia mentions decreased capacity to transport oxygen. Adding a qualifying phrase like “blood disorders” “not enough RBCs”, etc. might help.

Page 3 (Introduction), there is attempt to show the link between anemia and several factors such as age, sex, residency, initiation of complementary food, etc. Due to the fact that the citation was not contextual, the statement can be a misleading interpretation of the cited literature. On their own right, age, sex or residency cannot be linked to anemia. The preceding statements have already shown that globally a good percentage of children under 5 are anemic. Similarly, initiation of complementary food is an intervention to deal with under-nutrition, yet also linked to anemia. In this scenario, one would ask whether it is under-nutrition or initiation of complementary food that should be linked to anemia? In other words, the statement does not make it exclusive how age, sex & residency are linked with anemia. Please contextualize.

“Studies also indicated age and sex of the child, residency, under nutrition (stunting, wasting and thinness), early

initiation of complementary food, maternal health and education status, and low socioeconomic status were linked

with anemia (Belachew & Tewabe, 2020; Cardoso et al., 2012; Endris et al., 2022; Gebrie & Alebel, 2020; Ngesa &

Mwambi, 2014)”.

Results

It seems the data on hemoglobin levels has not been interrogated very well. No table is given, which could be useful . A simple categorization of children by age, sex, wealth status, infection status, maternal health status, etc, would probably give additional information. It would have also helped if data on height and body weight were available and analyzed so that nutritional status (including stunting, wasting, thinness) could be assessed. I would think this is a missed opportunity.

Page 8: How was it possible to verify if children had intestinal parasites or malaria? Was this documented? I am referring to the statement “Among those, 13(31%), 15(35.7%), 4(9.5%), and 37(88.1%) children had a history of malaria, diarrhea, intestinal parasites, and fever respectively.”

Page 10: As mentioned in my comments of the abstract section, food security status of families was assessed but not the increase in wealth score. Perhaps replacing the words “increasing” or “decreasing” may improve the sentence. See the quote below.

“The likelihood of anemia decreased by 0.95 (0.93, 0.97) with an increase in the age of the child in a month.

Similarly, a unit increase in wealth score decreased anemia prevalence with the odds of 0.86 (0.75, 0.99) among

children (Table.4).”

Discussion

1) There is a sudden mention of milk consumption in page 11. The logistic regression did not include data on frequency of food item consumption. This would have made it easier to understand why the authors suddenly mentioned milk consumption in page 11, in which it was indicated the study did not confirm the impact of milk confirmation on children’s iron status.

2) In Page 12 (1st paragraph) there is mention of contributions of infection/inflammation as being significant. It is unclear if this was a general statement or a reference to the study findings. As I understand it, neither malaria nor intestinal parasites were investigated by way of laboratory testing. By the same token, there were no measurements of iron, folate, vitamin B12 and vitamin A. The authors would need to convey their message in a different way to avoid confusions.

3) Overall, it seemed much effort was not exerted to collect as much relevant data as possible. The discussion is short. Either data is not available or not explored adequately.

English corrections (examples)

There are a lot of flaws in the use of English language (grammar, syntax, choice of words, etc..) that need to be addressed. Just some examples:

In the abstract, the phrase “were participated”, perhaps should just be “participated”

In Page 4, first sentence mentions “….. there is a huge difference between anemia prevalence and factors associated with anemia among studies conducted in different places (22)”. Here the comparison is about study sites (or the different studies), but the sentence seems to compare anemia prevalence with factors associated. This just needs differentiating the subjects of the sentence.

Page 4, last sentence of the introduction begins with “So, we would like to see the situation…. “. May be use phrases like “We sought to …….”.

Page 4 (last paragraph): Exclusion of severely ill children did make sense to a certain extent, but the impact is less clear as there are no biological measurements.

Page 5: Verb is missing in the sentence “The frequencies of consumption of over 67 locally available food ingredients for a week before the data was collected for the assessment of dietary diversity score.”

Page 6: the last phrase “was used to collect the data” in the sentence “The questionnaire was uploaded on KOBO collect survey, a mobile application that allows for collecting data using mobile devices, was used to collect the data.“ is repetition. It can be deleted.

Page 6: The paragraph on data quality assurance and control seems to require re-writing since it has repetitive phrases and statements. See below in the qoute:

“The questionnaire was prepared in English, translated into the local language (Amharic) and then translated back

into English to check consistency. Training was given to the data collectors on the study objective, requesting

consent, and data collection techniques for the data collectors. Then the data was collected using KOBO collect.

One of the investigators (SAB) closely monitored the data collection process with the server and onsite

supervision. Measurements were taken by following the recommended standards of the manufacturer. “

Page 6: The statement “Data was collected by using the KOBO collect” repeats several times.

Page 7: The phrase “there was no correlation between this independent variable” needs be explicit. It is unclear which variables are referred to. Furthermore, some sections of the same paragraph, unless it is re-written in a different way, seem to be a description of results, not expected in the M&M section. See the paragraph quoted below.

“Multicollinearity was assessed by examining the Variance Inflation Factor (VIF) and tolerance, which indicates that

there was no correlation between this independent variable. The final model was tested for its fitness by Hosmer

and Lemeshow p-value and the p-value was 0.879, which indicates the good fitness of the model.”

Page 11 (Discussion), please correct the phrase “in younger aged. ..”

Page 11 (Discussion), please correct the English in the phrase “For the most of the ….”.

Page 11 (Discussion), please correct the English in the phrase “for access adequate….”

Page 12 (1st paragraph): please correct the English in the phrase “non-nutrition related factors also contribution ……….”.

References

Journal names are given either in full designations or in abbreviated form. Be consistent.

6. PLOS authors have the option to publish the peer review history of their article (what does this mean?). If published, this will include your full peer review and any attached files.

**Do you want your identity to be public for this peer review?** For information about this choice, including consent withdrawal, please see our Privacy Policy.

Reviewer #1: No

Reviewer #2: No

---

## [Editor Report · Decision Letter 1]

8 Oct 2024

Prevalence of anemia and its associated factors among under-five children living in Arba Minch Health and Demographic Surveillance System Sites (HDSS), Southern Ethiopia

PGPH-D-24-01672R1

Dear Lecturer Bamboro,

We are pleased to inform you that your manuscript 'Prevalence of anemia and its associated factors among under-five children living in Arba Minch Health and Demographic Surveillance System Sites (HDSS), Southern Ethiopia' has been provisionally accepted for publication in PLOS Global Public Health.

Best regards,

Kamala Thriemer

Academic Editor